# Cardiorespiratory Fitness Mediates Cognitive Performance in Chronic Heart Failure Patients and Heart Transplant Recipients

**DOI:** 10.3390/ijerph17228591

**Published:** 2020-11-19

**Authors:** Florent Besnier, Béatrice Bérubé, Christine Gagnon, Miloudza Olmand, Paula Aver Bretanha Ribeiro, Anil Nigam, Martin Juneau, Lucie Blondeau, Michel White, Vincent Gremeaux, Louis Bherer, Mathieu Gayda

**Affiliations:** 1Preventive Medicine and Physical Activity Centre and Research Center, Montreal Heart Institute, Montreal, QC H1T 1N6, Canada; berubebeatrice@gmail.com (B.B.); christine.gagnon@gmail.com (C.G.); miloudza.olmand@umontreal.ca (M.O.); anil.nigam@icm-mhi.org (A.N.); martin.juneau@icm-mhi.org (M.J.); m_white@icm-mhi.com (M.W.); louis.bherer@umontreal.ca (L.B.); mathieu.gayda@icm-mhi.org (M.G.); 2Department of Medicine, Université de Montréal, Montreal, QC H3T 1J4, Canada; 3Research Centre, Institut Universitaire de Gériatrie de Montréal, Montreal, QC H3W 1W5, Canada; 4Research Centre, University of Montreal Hospital Research Centre, Montreal, QC H2X 0A9, Canada; paulaabribeiro@gmail.com; 5Montreal Health Innovations Coordinating Center, Montreal Heart Institute, Montreal, QC H1T 1C8, Canada; Lucie.Blondeau@mhicc.org; 6Sport Medicine Unit, Division of Physical Medicine and Rehabilitation, Swiss Olympic Medical Center, Lausanne University Hospital, 1011 Lausanne, Switzerland; Vincent.Gremeaux@chuv.ch

**Keywords:** heart failure, heart transplant, cardiorespiratory fitness, cognition

## Abstract

We compared cognitive profiles in chronic heart failure patients (HF), heart transplant recipients (HT) and healthy controls (HC) and examined the relationship between cardiorespiratory fitness ($\dot{V}$O_2peak_), peak cardiac output (CO_peak_) and cognitive performance. Stable HT patients (*n* = 11), HF patients (*n* = 11) and HC (*n* = 13) (61.5 ± 8.5 years) were recruited. Four cognitive composite scores targeting different cognitive functions were computed from neuropsychological tests: working memory, processing speed, executive functions and verbal memory. Processing speed and executive function scores were higher, which indicates lower performances in HF and HT compared to HC (*p* < 0.05). $\dot{V}$O_2peak_ and first ventilatory threshold (VT_1_) were lower in HF and HT vs. HC (*p* < 0.01). CO_peak_ was lower in HF vs. HT and HC (*p* < 0.01). Processing speed, executive function and verbal memory performances were correlated with $\dot{V}$O_2peak_, VT_1_ and peak cardiac hemodynamics (*p* < 0.05). Mediation analyses showed that $\dot{V}$O_2peak_ and VT_1_ mediated the relationship between group and processing speed and executive function performances in HF and HT. CO_peak_ fully mediated executive function and processing speed performances in HF only. $\dot{V}$O_2peak_ and CO_peak_ were related to cognitive performance in the entire sample. In addition, $\dot{V}$O_2peak_ and VT_1_ fully mediated the relationship between group and executive function and processing speed performances.

## 1. Introduction

Cognitive impairment (CI) affects up to 50% of patients with heart failure (HF) [1]. It is independently associated with mortality [2], poor quality of life, reduced functional capacity [3] and an overwhelming economic burden in Western countries [4,5]. HF-related CI is correlated with the severity of the disease [6] and mainly affects episodic memory, executive functions and processing speed [7]. Previous studies have shown that interventions durably treating HF, such as heart transplantation (HT), can improve cognition, for instance, memory, mental flexibility and attention [8,9,10]. Nevertheless, around 45% of HT recipients remain cognitively impaired [11,12,13], which is not trivial given that CI negatively affects therapeutic observance [14]. According to the “vascular dysfunction hypothesis” [7,15], one of the first mechanisms leading to CI is inadequate cerebral oxygenation and perfusion at rest, secondary to impaired cardiac output in HF and HT patients [9,16,17]. In line with this hypothesis, we previously showed that HT recipients had impaired cardiorespiratory fitness ($\dot{V}$O_2peak_), $\dot{V}$O_2_ at first ventilatory threshold (VT_1_), peak cardiac output (CO_peak_) and reduced cerebral oxygenation/perfusion during exercise compared to healthy controls [18]. In cardiac populations, $\dot{V}$O_2_ at VT_1_ corresponds to a moderate intensity, with aerobic metabolism as the main energy source. Importantly, VT_1_ is related to quality of life and daily physical activity without fatigue and/or dyspnea in cardiac patients. Moreover, in coronary heart disease (CHD) patients, we also showed that cognitive performance was related to $\dot{V}$O_2peak_, CO_peak_ and reduced cerebral oxygenation/perfusion [19]. In HT and HF patients, daily physical activity and/or functional capacity are related to cognitive performance [20,21,22,23]. Therefore, $\dot{V}$O_2peak_, VT_1_ and CO_peak_, reflecting cardiorespiratory fitness and functional capacity, could be important contributors/mediators of cognitive performance in these cardiac patients (HF–HT). To date, the relationship between cardiorespiratory fitness, CO_peak_ and cognitive performance in HF and HT has been poorly studied. Furthermore, it remains unclear whether these variables mediate cognitive performance in HF patients and HT recipients. In healthy older adults, our group showed that improvement in cardiorespiratory fitness mediates improvement in cognitive performance, with partial mediation observed in younger-old adults and full mediation in older-old adults aged over 75 years [24]. This suggests that cardiorespiratory fitness plays a crucial role in maintaining and improving cognitive functions in healthy older adults. This remains understudied in cardiac populations. The aims of this study are: (1) to compare cognitive profiles in HF patients, HT recipients and healthy controls (HC), and (2) to examine the relationship between cardiorespiratory fitness, peak cardiac output and their contribution (mediation) to cognitive performance in HF and HT patients compared to HC.

## 2. Materials and Methods

### 2.1. Participants

A total of 35 adults were recruited from the Cardiovascular Prevention and Rehabilitation Centre of the Montreal Heart Institute, including 11 chronic HF patients, 11 HT recipients and 13 HC. This study is part of a project conducted at the Montreal Heart Institute and approved by its research ethics board (ClinicalTrials.gov Identifier: NCT03018561). All participants provided written informed consent prior to inclusion [18]. Only subjects who completed the neurocognitive evaluation were included in the sample. Detailed inclusion and exclusion criteria for HC, HT recipients and HF patients are shown in Appendix A.

### 2.2. Study Design

All participants underwent a baseline evaluation that included a medical history and physical examination with measurements of height and body mass. Main components of cognition were assessed (working memory, executive functions, processing speed and verbal memory) using a comprehensive neuropsychological test battery. Participants performed a maximal cardiopulmonary exercise test with cardiac hemodynamics (impedance cardiography) and ECG measurements (see below for further details) [18,25].

### 2.3. Maximal Cardiopulmonary Exercise Test (CPET)

CPET was performed on an ergocycle (Ergoline 800S, Bitz, Germany), with an individualized protocol that included a 3-min warm up at 20 watts, followed by a power increase of 10 to 20 watts/min until exhaustion at a pedaling speed of >60 rpm [25]. Gas exchange [dioxygen ($\dot{V}$O_2_) and carbon dioxide ($\dot{V}$CO_2_)] was measured breath-by-breath continuously at rest, during exercise and at recovery using a metabolic gas analyzer system (Oxycon Pro, Jaegger, Germany) and then was averaged every 15 s for analysis. $\dot{V}$O_2_ at first ventilatory threshold (VT_1_) was also calculated as previously published [18]. The highest $\dot{V}$O_2_ value (15 s averaged) reached during the exercise phase was considered as the $\dot{V}$O_2peak_, and peak power output (PPO) was defined as the workload reached at the last fully completed stage [18]. The electrocardiogram (ECG) was continuously monitored (Marquette, case 12, GE Healthcare) during the test. Blood pressure (manual sphygmomanometer: Welch Allyn Inc., Chicago, IL, USA) and rate of perceived exertion (RPE) were measured every 2 min throughout the test. During CPET, cardiac hemodynamics (cardiac output: CO, cardiac index: CI, and left cardiac work indexed: LCWi) were measured continuously at rest, during exercise and a recovery using an impedance cardiography device as previously described (PhysioFlow^®^, Enduro model, Manatec, France) [18,25].

### 2.4. Cognitive Evaluation

The following neuropsychological tests were administered in a fixed order. Digit Span (DS) is administered by the recall of forward and backward digit sequences. The forward span assesses auditory short-term memory, whereas the backward span targets working memory. The Rey auditory verbal learning test (RAVLT) is used to assess episodic memory in auditory and verbal domains and learning [26] through 5 learning trials of a 15-word list. In this test, participants must recall as many words as possible immediately after each trial of the learning phase, after an interfering list, as well as following a 30-min delay. The Digit Symbol Substitution Test (DSST) assesses processing speed. Participants have to associate symbols with numbers referring to a response key as fast as possible for 120 s. The Stroop Color-Word Interference Test (SCWIT) contains 4 different conditions (naming, reading, inhibition, switching). In the naming condition, participants name the color of rectangles. In the reading condition, participants read color words printed in black ink. In the inhibition condition, participants inhibit reading in order to name the incongruent ink colors in which the words are printed (e.g., RED printed in green ink). In the switching condition, participants are asked to alternate between inhibition and word reading. The first two conditions tap processing speed. The third condition assesses inhibition, and the fourth, flexibility and switching, both mechanisms of executive functioning [26]. The Trail Making Test (TMT) includes part A (TMT A), which measures processing speed and visuospatial abilities, and part B (TMT B), which assesses attentional control and cognitive flexibility. In part A, participants must link numbers 1–25 in ascending order as fast as possible. In part B, participants must alternate between letters and numbers, linking number–letter sequences in an ascending and alphabetic order, here again as fast as possible (e.g., 1-A-2-B-3-C, etc.). All cognitive scores were first transformed into standardized Z scores (Z score = (value – mean value of all the subjects)/standard deviation). Then, four composite cognitive scores were calculated using raw Z scores as follows [27]: (1) working memory = ((DS forward + DS backward scores)/2); (2) processing speed = ((DSST+ TMT A+ Stroop 1+ Stroop 2 scores)/4); (3) executive functioning = ((Trail B+ Stroop 3+ Stroop 4 scores)/3); and (4) verbal memory/episodic memory (immediate recall + delayed recall + total words scores recalled during the 5 learning trials from the RAVLT test/3). Cronbach’s alphas (α) were used to verify the internal consistency between all measures included in a composite score, considering a Cronbach α of > 0.7 as acceptable (see Results section) [27].

### 2.5. Statistical Analysis

Data were summarized by mean ± standard deviation (SD). CPET, cardiac hemodynamics at peak exercise and cognitive performances were compared between groups using one-way ANOVAs. In the case of a significant main group effect, pairwise comparisons were used to determine which group differences were significant. Correlations between cognitive composite scores, $\dot{V}$O_2peak_, VT_1_ and cardiac hemodynamics at peak exercise were assessed with a Pearson coefficient (R). Mediation analyses were then conducted according to the methodology developed by Hayes et al. [28]. Mediation analyses were used to determine if $\dot{V}$O_2peak_, VT_1_ and CO_peak_ significantly influenced the relationship between group (HT, HF, HC) and cognitive performance. The direct effect of group on cognition and the indirect effect of group running through $\dot{V}$O_2peak,_ VT_1_ and peak cardiac output were analyzed. The direct path *a_n_* represents the regression coefficient for the dummy-coded independent variable (IV) when the mediator variable (MV) is regressed on the independent variable (IV), while *b* is the direct path coefficient for the MV when the DV is regressed on the MV and IV. The product of the coefficient method was used to compute the indirect effect. This method determines the indirect effect by multiplying the regression coefficients: a_n_ × b = a_n_b. The total effect (c) is the direct effect + the indirect effect. In other words, the relationship between the IV and the DV is decomposed into a direct link and an indirect link. A direct effect (c’) refers to the relationship between the IV and the DV after controlling for M. $\dot{V}$O_2peak_ was included as the MV between the categorical IV “group” (HC, HF, HT, where HF and HT will be compared to HC) and the DV composite cognitive scores. The same procedure was applied with VT_1_ and peak cardiac output. The significance of the indirect effects was tested using bias-corrected bootstrap confidence intervals (based on 5000 replications). Confidence intervals that did not contain zero represented significant effects. Partial mediation was denoted if the direct effect and the indirect effect were significant. Full mediation was denoted if a non-significant direct effect was associated with a significant indirect effect. All statistical tests were two-sided and conducted at a 0.05 significance level. Statistical analyses were performed with the use of SAS software, version 9.4 (SAS Institute) except for the mediation analyses that were conducted using Stata SE 15.1 (StataCorp LP, College Station, TX, USA) according to the methodology developed by Hayes et al. [28].

## 3. Results

### 3.1. Group Comparisons

Baseline clinical and sociodemographic characteristics were similar for the three groups (HC, HF, HT) and are presented in Table 1.

For CPET parameters, compared to HC, HF and HT had significant lower values for all of the parameters (Table 2). $\dot{V}$O_2peak_ in HF and HT was significantly reduced compared to HC (17.6 vs. 26.5 vs. 37.7 mL/min/kg, respectively, *p* < 0.001). At peak exercise, cardiac output (CO_peak_), cardiac index (CI_peak_) and left cardiac work index (LCWI_peak_) significantly differed according to group (Table 2), with lower values seen in HF vs. HT and HC but with no statistical difference between HT and HC.

Table 3 shows the neuropsychological tests and the composite cognitive scores for participants in the three groups. Cronbach’s alphas (α) were 0.640 for working memory, 0.787 for processing speed, 0.789 for executive functioning and 0.918 for verbal memory. The HF and HT groups demonstrated significantly higher composite z scores, suggesting reduced performances for processing speed and executive function when compared to HC participants (*p* < 0.05). Working memory performance was similar in the three groups (*p* = 0.425). The verbal memory composite score was greater in HC compared to HF (*p* < 0.001) but was similar compared to HT (*p* = 0.178) (Table 3).

### 3.2. Univariate Correlations

Univariate correlations are detailed in Appendix A. Executive functioning, processing speed and verbal memory were correlated with $\dot{V}$O_2peak_, VT_1_, peak power output, oxygen pulse, HRpeak, CO_peak_, CI_peak_ and LWCi_peak_ (*p* < 0.05) in the whole sample.

### 3.3. Mediation Analyses

Figure 1 presents the results of the mediation analyses. The direct effect of group predicts $\dot{V}$O_2peak_ (a_1_ path coefficient for HF = −20.1, *p* < 0.001, and a_2_ path coefficient for HT = −11.2, *p* < 0.001). Higher $\dot{V}$O_2peak_ was significantly associated with faster responses (reflected by a lower value of executive functioning Z score) (b = −0.046, *p* < 0.001). Each of the indirect effects of group on executive functioning through $\dot{V}$O_2peak_ was significant (a_1_b = 0.92, *p* = 0.001, and a_2_b = 0.51, *p* = 0.009) as well as the total indirect effect (a_1_b +a_2_b = 1.43; *p* = 0.002; 95% IC: 0.541; 2.329). In contrast to the control group (HC), HF and HT had executive functioning scores that were increased by 0.92 and 0.51 units, respectively (unfavorable because of slower responses). A lower $\dot{V}$O_2peak_ (from the sign of a_1_ and a_b_) increased the executive functioning Z score (from the sign of b), meaning that the time to complete the task increased.

The direct effect (c′ = c′_1_ + c′_2_) of group on executive functioning is 0.55 (*p* = 0.310) and the proportion of total effect that is mediated is 0.72 (a_n_b/c_n_), meaning that 72% of the relationship between group and executive functioning performances is indirect via $\dot{V}$O_2peak_. This is consistent with a full mediation of the relationship between group and executive functioning through $\dot{V}$O_2peak_, where $\dot{V}$O_2peak_ is responsible for 72% of the effect of group difference on executive function performance. Mediation analyses showed that $\dot{V}$O_2peak_ and $\dot{V}$O_2_ at first ventilatory threshold (VT_1_: Appendix A) fully mediate the relationship between group and executive function and processing speed performances, but not verbal memory performance. In addition, CO_peak_ fully predicts executive function and processing speed performances in HF only (Appendix A).

## 4. Discussion

In this cross-sectional study, we first compared cognitive profiles of HF and HT patients to those of healthy controls. We also examined the relationship between cardiorespiratory fitness, peak cardiac output and their mediation on cognitive performance in HF and HT patients compared to HC. The main results of our study can be summarized as follows: (1) In addition to a reduced $\dot{V}$O_2peak_, VT_1_ and CO_peak_, HF and HT have reduced cognitive performance (for processing speed and executive function composite scores) compared to HC. (2) Cardiorespiratory fitness and cardiac hemodynamics at peak exercise were associated with cognitive function (univariate correlation). (3) Cardiorespiratory fitness ($\dot{V}$O_2peak_, VT_1_) fully mediates the relationship between group and cognitive performance in HF and HT patients and peak cardiac output fully mediates cognitive performance in HF patients only.

### 4.1. Cognition in Chronic Heart Failure and in Heart Transplant Recipients

Cognitive impairment (CI) has been documented in chronic HF patients shortly (four months) after heart transplantation and in the long term in stable HT patients (one to 16 years) after their transplantation [13,16,29].

The cognitive domains affected in cardiac populations are mainly executive function, processing speed and memory [12,13,30,31,32,33]. Interestingly, our results showed lower scores for executive functions, verbal memory and processing speed composite scores in HF compared to HC. HT recipients obtained higher scores compared to HF in some cognitive functions, namely verbal memory, executive function and processing speed, but still performed lower compared to HC for the two later domains. Indeed, verbal memory performance was similar between HC and HT groups. Overall, our results suggest that multiple cognitive domains are not totally recovered even in stable HT recipients. Importantly, executive functions, which remained lower in HT recipients compared to HC, are related to medication adherence and instrumental activities of daily living, such as housework, preparing meals and engaging in physical activities. Consequently, deficits in executive functions can lead to difficulty in performing health self-management [20,21].

### 4.2. Related Mechanisms of Cognitive Impairment

HT recipients are more likely to present CI as a consequence of previous micro- and macrovascular disease [34], chronotropic incompetence [35], immunosuppressant therapy that could be linked to brain function [36] and even micro-cerebrovascular events (e.g., embolism) or lower hemoglobin concentration [37]. Moreover, most of the HT recipients were previously end-stage HF patients, who are at high risk of developing CI. Indeed, CI in HF is hypothesized to be due, at least in part, to a chronic reduction in cerebrovascular perfusion and oxygenation, and to oxidative stress and a pro-inflammatory status, which would then lead to structural changes in the brain [23,38,39,40]. Importantly, CI in HF patients has been related to the poorest NYHA (New York Heart Association) class, time of HF diagnostic, left ventricular ejection fraction (LVEF) under 30% and resting cardiac function [9,16], but there is less evidence exploring CI in long-term HT survivors [13].

In a pre- and post-heart transplant study with end-stage HF patients, cognitive scores (memory and dexterity/coordination and memory) were correlated with resting cardiac index (r = 0.32 and 0.52, respectively) [9]. The effect of HT on cognition remains unclear and needs more follow-up evaluations to elucidate the underlying mechanisms, as well as the effects of immunosuppressive treatments [29]. For instance, Grimm et al. observed that after HT, global cognitive performance was improved at four months, compared to prior HT, but then declined at 12 months. Cumulative cyclosporine dosage (immunosuppressive drug) was the only independent predictor of individual cognitive brain function after transplantation, meaning that this treatment could have a long-term negative impact on cognition [29].

### 4.3. Cardiorespiratory Fitness in Chronic Heart Failure and in Heart Transplant Recipients

Our results are in line with previous studies assessing cardiorespiratory fitness and CO at peak exercise with cardiac bioimpedance in HT and HF [18,41]. In chronic HF, exercise intolerance is multifactorial. The “systemic” pathophysiology involves central cardiac dysfunction but also peripheral abnormalities in the skeletal muscle, neuro-hormonal, endothelial and biochemical functions (oxidative stress and exacerbated pro-inflammatory status), aggravating myopathy and deconditioning [42]. In HT recipients, impaired cardiorespiratory fitness and cardiac hemodynamics at peak exercise are the result of both central (cardiac denervation, diastolic dysfunction) and peripheral abnormalities (vascular dysfunction, reduced skeletal muscle oxidative fibers, enzymes, capillarity) that limit O_2_ delivery and extraction by the exercising skeletal muscles [43]. Cardiorespiratory fitness in HT recipients is a strong predictor of long-term post-transplantation survival [44]. Accordingly, both for HF and HT, exercise training in order to improve $\dot{V}$O_2peak_ is part of cardiac rehabilitation guidelines [45].

### 4.4. Relationship between $\dot{V}$O_2peak_, CO_peak_ and Cognitive Performance

Resting cardiac hemodynamic function, such as cardiac output, is also related to cognitive functions in HF and HT patients [9,17,46]. Cardiorespiratory fitness is also well correlated with cognitive performance (measured at rest) in healthy adults [15]. A CPET test allows an integrated analysis of respiratory, cardiac, vascular and muscular adaptations (Fick equation: $\dot{V}$O_2_ = CO × (A-V) O_2_). Our study demonstrated that cardiorespiratory fitness ($\dot{V}$O_2peak_) and $\dot{V}$O_2_ at first ventilatory threshold correlated and fully mediated the relationship with processing speed and executive function performance, but not with verbal memory scores. It is well described in the literature that cardiorespiratory fitness is related to performances in global cognition, executive function, processing speed, attention and driving performance in HF patients [15,22,27,47], as well as with white matter structure in elderly adults [48]. This association is mediated, in part, by increases in brain perfusion and the ability of cerebral blood vessels to respond to demand [7]. Compared to HC, HT recipients had reduced $\dot{V}$O_2peak_, cardiac function and cerebral oxygenation–perfusion during exercise [18]. However, cognition was not measured in this study. In CHD patients, $\dot{V}$O_2peak_ and CI_max_ were both related to exercise-related cerebral oxygenation–perfusion and cognitive function (*p* < 0.005) [19]. In line with the “vascular dysfunction hypothesis” [49], CI may be due to a dysfunction of the mechanism involved in the regulation of cerebral blood flow and responsible for pathophysiological consequences of cerebral microvascular dysregulation (cerebral autoregulation/myogenic constriction, endothelium-dependent vasomotor function, neurovascular coupling responses). Abnormal cerebral autoregulation and cerebral vasoreactivity, potentiating the risk of impaired cerebral perfusion, therefore increasing the occurrence of silent brain damage and microvascular injury, are frequent in HF [50], but remain under discussion in HT patients [51,52]. These cerebrovascular phenomena can lead to the development of premature CI [49]. Regular physical activity improves cardiorespiratory fitness in both HF and HT and can have beneficial impacts on neurovascular functions and brain autoregulation in HF [53,54]. However, there are no data available regarding HT recipients. These benefits on cerebrovascular function may explain the positive impact of exercise on cognition, including benefits on learning, memory, attention and executive functions in healthy adults [55,56] and HF patients [38,57,58]. The positive beneficial effect of exercise training on cognitive function is likely due to its pro-neurogenic effects [59]. Some evidence also revealed that regular exercise may increase angiogenesis, neurogenesis, synaptogenesis and neurotransmitter synthesis in cerebral structures, as well as increasing grey and white matter volume [35,55]. The increased cerebral flow, especially after transplant [60], could stimulate neurobiological mechanisms leading to an improvement in cognitive function. Longitudinal studies exploring the effects of exercise training on cardiorespiratory fitness, cognition and cerebrovascular function in end-stage HF and HT are thus needed.

### 4.5. Study limitations and Research Perspectives

Our study has limitations that should be highlighted. First, the small number of participants in each group and the fact that they were recruited in a single center induces a potential recruitment bias. Larger cross-sectional and longitudinal studies before/after heart transplant with medium to long-term follow-ups are needed, as well as groups involved in exercise training programs. Furthermore, larger groups with different etiologies of HF leading to HT need to be taken into account in the analyses. Moreover, physical activity/sports and level of activities of daily living were not evaluated in a structured way in the study protocol, yet cognitive performances could be impacted by physical activity level, independently of cardiorespiratory fitness [21]. Furthermore, the potential positive effects of medication such as angiotensin II receptor blockers and angiotensin-converting enzyme inhibitors on cognitive performances in HF and HT patients cannot be excluded [61]. Moreover, because of the very low proportion of women in our sample, we were unable to discuss sex-related differences. Yet according to Lee et al. [62], older women with HF had higher CI (15%) and more inadequate health literacy (56.7%) compared to men. In their study, cognitive function was the strongest predictor of health literacy in men (β = 3.668, *p* < 0.001) and women (β = 2.926, *p* = 0.004). Future studies investigating sex-related differences in cardiac disease-related CI and the underlying pathophysiological mechanisms are needed. Finally, we did not differentiate in our analyses HT patients presenting evidence of cardiac reinnervation (according to heart rate responses to exercise testing and recovery). Partial reinnervation is found in some HT recipients and involves the myocardial muscle, the sinoatrial node and the coronary vessels, but remains incomplete and regionally limited many years post-transplant. Restoration of cardiac innervation can improve cardiorespiratory fitness as well as blood flow regulation in the coronary arteries, and may positively affect the heart–brain axis, the regulation of cerebral circulation and thus of cognitive functions. However, further studies are needed to investigate this hypothesis.

## 5. Conclusions

Our results demonstrate that cognitive performance is reduced in HF and HT patients compared to HC, for processing speed and executive function domains. For verbal memory, HT participants perform better than HF, and perform comparably to HC. Moreover, $\dot{V}$O_2peak_, VT_1_ and cardiac output at peak exercise were related to cognitive performances (processing speed, executive function and verbal memory). The relationship between group (HF and HT vs. HC), executive functioning and processing speed was fully mediated by cardiorespiratory fitness ($\dot{V}$O_2peak_) and $\dot{V}$O_2_ at VT_1_. Finally, peak cardiac output fully mediated processing speed and executive performances in HF only.

## Figures and Tables

**Figure 1 ijerph-17-08591-f001:**
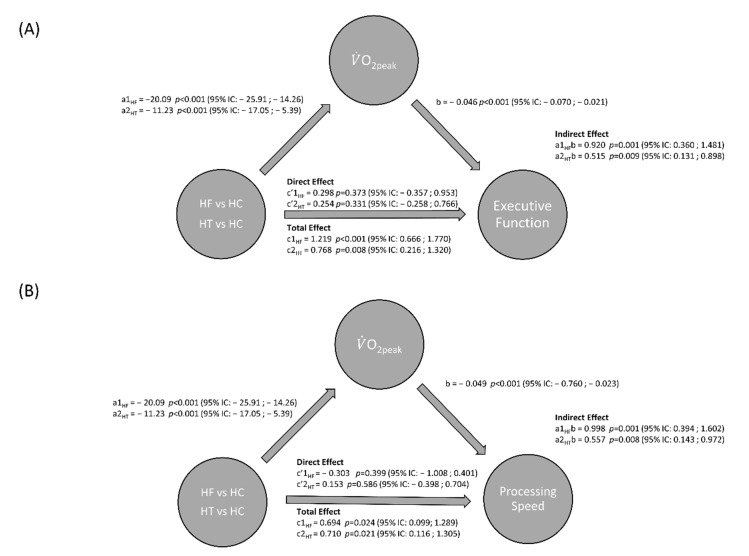
The mediating effect of cardiorespiratory fitness ($\dot{V}$O_2peak_) on the relationship between group and executive function (**A**) and processing speed (**B**). The categorical variable group has three levels: healthy controls are coded as 0, heart failure as 1 and heart transplant as 2. The healthy control group is the reference group. a_n_ path is the effect of group on $\dot{V}$O_2peak_ and b path is the effect of $\dot{V}$O_2peak_ on executive functioning. c_n_ is the total effect of group on executive functioning and c’_n_ is the direct effect of group on executive functioning.

**Table 1 ijerph-17-08591-t001:** Baseline clinical and sociodemographic characteristics in healthy controls (HC), heart failure patients (HF) and heart transplant recipient (HT) participants.

	HF *n* = 11	HT *n* = 11	HC *n* = 13	*p* Value
	Mean ± SD	Mean ± SD	Mean ± SD	
Age (years)	66.18 ± 7.88	58.55 ± 7.92	60.15 ± 8.49	0.081
Sex (F/M)	1/10	1/10	1/12	-
Heigh (cm)	172.45 ± 8.26	169.73 ± 7.36	172.69 ± 8.28	0.621
Body mass (kg)	77.38 ± 10.91	78.07 ± 15.85	73.53 ± 8.09	0.532
BMI (kg/m^2^)	25.97 ± 2.89	27.04 ± 4.95	24.65 ± 2.03	0.236
Body surface (m^2^)	1.92 ± 0.17	1.91 ± 0.21	1.88 ± 0.14	0.796
Level of education (years)	14.36 ± 3.83	11.82 ± 3.09	14.67 ± 2.67	0.086
Time from transplantation (years)	-	7.40 ± 5.40	-	-
Medication				
Immunosuppressive drugs *		11 (100%)		
Beta blockers	11 (100%)	3 (27%)		
ACE inhibitors	6 (55%)	0		
ARBs	6 (55%)	6 (55%)		
Diuretics	9 (82%)			

BMI: body mass index, ACE: angiotensin I-converting enzyme, ARBs: angiotensin receptor blockers, * Rapamycine, Myfortic, Prograf, Cyclosporine, Cellecept, Tacrolimus.

**Table 2 ijerph-17-08591-t002:** Cardiopulmonary exercise test and cardiac hemodynamic variables in HC, HF and HT participants.

Cardiopulmonary and Hemodynamic Variables	HF *n* = 11	HT *n* = 11	HC *n* = 13	*p* Value	HF vs. HT	HF vs. HC	HT vs. HC
	Mean ± SD	Mean ± SD	Mean ± SD				
Peak power output (watts)	87.27 ± 26.96	134.09 ± 66.96	225.77 ± 54.42	<0.0001	0.051	<0.0001	0.002
$\dot{V}$O_2peak_ (mL/min)	1346 ± 316	1981 ± 626	2780 ± 606	<0.0001	0.009	<0.0001	0.005
$\dot{V}$O_2peak_ (mL/min/kg)	17.60 ± 4.31	26.46 ± 10.12	37.69 ± 7.27	<0.0001	0.019	<0.0001	0.007
RER	1.14 ± 0.09	1.12 ± 0.09	1.16 ± 0.08	0.480	-	-	-
HR_peak_ (bpm)	113.91 ± 20.81	138.27 ± 21.28	157.15 ± 11.47	<0.0001	0.013	<0.0001	0.019
O_2_ pulse (mLO_2_/bpm)	12.02 ± 2.40	14.17 ± 3.15	17.70 ± 3.72	0.001	0.122	<0.001	0.011
SBP_peak_ (mmHg)	141.4 ± 19.7	176.8 ± 22.6	184.8 ± 23.8	<0.0001	0.001	<0.0001	0.384
DBP_peak_ (mmHg)	68.5 ± 6.3	74.1 ± 7.7	78.6 ± 11.6	0.032	0.076	0.014	0.267
CI_peak_ (L/min/m^2^)	5.35 ± 1.50	6.97 ± 1.52	7.98 ± 1.05	<0.001	0.008	<0.0001	0.080
CO_peak_ (L/min)	10.29 ± 2.89	13.11 ± 2.57	14.51 ± 2.13	0.001	0.013	<0.001	0.185
LCWi_peak_ (kg.m/m^2^)	6.50 ± 2.55	9.85 ± 2.42	11.38 ± 2.28	<0.0001	0.003	<0.0001	0.131
$\dot{V}$O_2_ at VT_1_ (mL/min)	918 ± 170	1309 ± 284	2108 ± 528	<0.0001	0.002	<0.0001	<0.001
Power at VT_1_ (watts)	51.36 ± 17.62	69.90 ± 27.20	162.08 ± 47.79	<0.0001	0.086	<0.0001	<0.0001

VT_1_: first ventilatory threshold. RER: respiratory exchange ratio. HR_peak_: heart rate at peak exercise. SBP/DBP: systolic and diastolic blood pressure. CI: cardiac index. CO: cardiac output. LCWi: left cardiac workout index.

**Table 3 ijerph-17-08591-t003:** Neuropsychological variables in HC, HF and HT participants.

	HF *n* = 11	HT *n* = 11	HC *n* = 13	*p* Value	HF vs. HT	HF vs. HC	HT vs. HC
Neuropsychological Tests	Mean ± SD	Mean ± SD	Mean ± SD				
MMSE	26.82 ± 1.08	27.09 ± 2.47	28.85 ± 0.90	<0.001	0.742	<0.0001	0.045
Geriatric Depression Scale	9.18 ± 8.61	4.36 ± 2.25	3.33 ± 3.28	0.140	-	-	-
Forward	8.91 ± 2.39	9.55 ± 1.75	10.00 ± 1.21	0.396	-	-	-
Backward	6.45 ± 2.38	6.36 ± 1.75	7.17 ± 2.29	0.622	-	-	-
DSST	52.45 ± 12.14	51.64 ± 14.40	69.62 ± 12.15	0.002	0.883	0.003	0.002
TMT A (s)	47.53 ± 15.54	38.86 ± 10.81	36.34 ± 12.59	0.114	-	-	-
TMT B (s)	112.10 ± 54.53	85.21 ± 34.30	67.46 ± 17.98	0.043	0.190	0.024	0.161
TMT (B-A)/A	1.33 ± 0.92	1.16 ± 0.64	0.95 ± 0.44	0.397	-	-	-
**Stroop Test**							
Stroop 1 (s)	31.19 ± 4.09	33.89 ± 5.67	29.41 ± 4.96	0.135	-	-	-
Stroop 2 (s)	22.63 ± 4.57	23.39 ± 4.33	20.10 ± 3.35	0.144	-	-	-
Stroop 3 (s)	72.39 ± 9.60	63.28 ± 9.19	52.05 ± 11.02	<0.001	0.041	<0.0001	0.011
Stroop 4 (s)	74.21 ± 13.51	73.22 ± 17.76	55.86 ± 16.18	0.014	0.884	0.010	0.014
**RAVL Test**							
Immediate recall	6.55 ± 2.25	8.18 ± 4.29	10.38 ± 3.04	0.009	0.280	0.002	0.171
Delayed recall	6.40 ± 2.32	9.45 ± 3.42	10.00 ± 2.37	0.013	0.017	0.006	0.647
Rey 1–5 total words	35.18 ± 5.46	46.82 ± 8.65	51.62 ± 8.11	<0.0001	0.001	<0.0001	0.132
**Composite Z scores**							
Working memory	−0.21 ± 1.11	−0.06 ± 0.82	0.25 ± 0.60	0.425	-	-	-
Processing speed	0.26 ± 0.61	0.28 ± 0.78	−0.43 ± 0.73	0.029	0.958	0.024	0.021
Executive functioning	0.59 ± 0.63	0.14 ± 0.75	−0.63 ± 0.60	<0.001	0.120	<0.0001	0.008
Verbal memory	−0.78 ± 0.53	0.12 ± 0.99	0.56 ± 0.78	0.001	0.012	<0.001	0.178

MMSE: Mini-Mental State Examination. DSST: Digit Symbol Substitution Test. TMT: Trail Making Test. RAVL: Rey auditory verbal learning test.

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
