# Peer review of "Cardiorespiratory Fitness Mediates Cognitive Performance in Chronic Heart Failure Patients and Heart Transplant Recipients"

_ijerph, 2020, doi:10.3390/ijerph17228591_

Round 1

Reviewer 1 Report

This manuscript is a really interesting exploration of the interactions between cardiac health and cognitive function that builds on previous research from the same group of authors. I think it is a valuable addition to the literature, but have some suggested edits:

Line 69-70: What is the difference, if any, between HT patients and HT recipients? Please clarify

Line 142: Were post-hoc corrections used with the ANOVAs? If yes, please provide that information. If not, please use such corrections

Lines 148-162: I think this discussion of the technicalities of the mediation analysis may be able to be trimmed down. What processes, if any, were used in Stata for the mediation analysis? As it is written, it sounds like the coefficients were calculated by hand.

Table 1: Please change “Years of studies” to “years of schooling completed” or similar

Table 1: Please address any possible role for the various medications in explaining any cognitive impairment in the Discussion or Limitations

Figure 1: Please relabel a1hf, a2ht, c’1hf, etc. for clarity.

Lines 253 and 254: Please explain what NYHA and LVEF are

Please also proofread the manuscript for typos.

Author Response

Comments and Suggestions for Authors

This manuscript is a really interesting exploration of the interactions between cardiac health and cognitive function that builds on previous research from the same group of authors. I think it is a valuable addition to the literature, but have some suggested edits:

We kindly thank Reviewer 1 for the time taken to revise this manuscript in order to improve it. We hope that the revised version will be acceptable for publication. 

Line 69-70: What is the difference, if any, between HT patients and HT recipients? Please clarify

We apologize for the typo, which was corrected: HT has been corrected for HF, meaning heart failure patients.

Line 142: Were post-hoc corrections used with the ANOVAs? If yes, please provide that information. If not, please use such corrections

Thank you for reminding us of this oversight. The following sentence was added, “In the case of a significant main group effect, pairwise comparisons were used to determine which group differences were significant”

Lines 148-162: I think this discussion of the technicalities of the mediation analysis may be able to be trimmed down. What processes, if any, were used in Stata for the mediation analysis? As it is written, it sounds like the coefficients were calculated by hand.

Thank you for this comment. All the procedures were done with Stata, not by hand. Please see the methodological article in the reference section #28: Hayes AF, Preacher KJ. Statistical mediation analysis with a multicategorical independent variable. Br J Math Stat Psychol. 2014;67:451-470. In this paper, the authors describe how to run this analysis with Stata; we used exactly the same methodology. We added the reference at the end of this paragraph. Furthermore, the paragraph was trimmed down by 2-3 lines.

Table 1: Please change “Years of studies” to “years of schooling completed” or similar

 This was modified for: Level of education (years)

Table 1: Please address any possible role for the various medications in explaining any cognitive impairment in the Discussion or Limitations

We thank Reviewer 1 for this comment. Several meta-analyses (e.g., Rouch L et al. CNS Drugs. 2015 Feb;29(2):113-30, Stuhec M et al. Eur Psychiatry. 2017 Oct;46:1-15) have shown that ARBs and ACE can have positive impacts on cognitive function and cognitive decline in adults. The potential influence of pharmacological treatments, particularly in the HF group, was added in the limitation section (4.5).

Figure 1: Please relabel a1hf, a2ht, c’1hf, etc. for clarity.

The nomenclature a, b, c, c’ is the common way to report the direct effect, indirect effect and total effect. We will keep this nomenclature regarding the mediation analysis. This was validated by our statistical experts service at the Montreal Heart Institute Coordinating Center.

Lines 253 and 254: Please explain what NYHA and LVEF are

Abbreviations were defined as: NYHA - New York Heart Association, LVEF – left ventricular ejection fraction.

 Please also proofread the manuscript for typos.

The manuscript was proofread for typos and English.

Reviewer 2 Report

Major comments 

  1. HF patients and HT recipients had pharmacological treatments (BBs, ACE inhibitors, ARBs, diuretics) during the study. The authors should discuss the impact of these drugs on cognitive function. 
  2. In the protocol, only a woman was included. The above represent the 7-9% of each group. Would be interesting to include more women in the analysis. Is this being not possible, at least, the authors should discuss the influence of gender in the results obtained. 

Minor comments 

  1. In lines 304-305, the words “exercise training” and in line 315 “cross-sectional” are bigger than the rest, change the size font 

Author Response

Comments and Suggestions for Authors

We kindly thank Reviewer 2 for the time taken to revise this manuscript in order to improve it. We hope that the revised version will be acceptable for publication. 

Major comments 

  1. HF patients and HT recipients had pharmacological treatments (BBs, ACE inhibitors, ARBs, diuretics) during the study. The authors should discuss the impact of these drugs on cognitive function. 

We thank Reviewer 2 for this comment. Several meta-analyses (e.g., Rouch L et al. CNS Drugs. 2015 Feb;29(2):113-30, Stuhec M et al. Eur Psychiatry. 2017 Oct;46:1-15) have shown that ARBs and ACE can have positive impacts on cognitive function and cognitive decline in adults. The potential influence of pharmacological treatments, particularly in the HF group, was added in the limitation section (4.5).

  1. In the protocol, only a woman was included. The above represent the 7-9% of each group. Would be interesting to include more women in the analysis. Is this being not possible, at least, the authors should discuss the influence of gender in the results obtained. 

We thank Reviewer 2 for the relevant comments. Unfortunately, we are not able to add more women to our sample. Because of our quite low sample size and the few of women in our sample (1 women in each group), we are not able to discuss any influence of sex on the results.

In chronic heart failure literature, data on sex-related differences in cognitive performance are limited; Pressler et al. reported poorer scores in memory, psychomotor speed and visuospatial recall in men than women, despite men having less severe HF (Pressler S.J. et al. J Card Fail, 2010.) Rochette et al. also observed that women outperform men in memory, but not in attention and executive function tests (Rochette A.D. et al. J Cardiovasc Nurs 2017). Recently Lee et al. IJERPH 2017, reported that older women with HF had higher cognitive impairment (15%) and inadequate health literacy (56.7%) compared to men. In their study, cognitive function was the strongest predictor of health literacy in men (β = 3.668, p < 0.001) and women (β = 2.926, p = 0.004). The exact pathophysiological explanation of these sex-related differences are still under investigation, however, one hypothesis suggests sex as a potential moderator of the effects of cerebral autoregulation on cognitive performances. To address this issue, we added a short paragraph in the limitation section.

Minor comments 

  1. In lines 304-305, the words “exercise training” and in line 315 “cross-sectional” are bigger than the rest, change the size font 

Corrections done.
